# Prevalence of diabetic retinopathy in children and young people living with diabetes: protocol for a systematic review

Maria Carolina Ibanez-Bruron,[1,2] Ameenat L Solebo,[1,2,3,4,5] Phillippa M Cumberland,[1,2] Jugnoo S Rahi[1,2,3,4,5]

[1]GOS Institute of Child Health, University College London, London, UK
[2]Ulverscroft Vision Research Group, London, UK
[3]Great Ormond Street Hospital for Children NHS Foundation Trust, London, UK
[4]Moorfields Eye Hospital NHS Foundation Trust, London, UK
[5]Institute of Ophthalmology, University College London, London, UK

**Correspondence to**
Professor Jugnoo S Rahi;
j.rahi@ucl.ac.uk

## ABSTRACT

**Introduction** The frequency of diabetes mellitus in childhood is increasing. Thus, more children and young people are at risk of developing diabetic retinopathy and diabetes related visual impairment. However, there is no consensus on optimal screening strategies for the paediatric population reflecting the lack of clarity about the current burden of disease in this group. We aim to estimate the prevalence of diabetic retinopathy in children and young people living with types 1 or 2 diabetes, and to investigate potential sources of heterogeneity in this figure so as to inform screening strategies for this population.

**Methods and analysis** PubMed and EMBASE will be searched from 1995 to 2016 using the OvidSP platform with no language restriction. Additionally, manual review of the references lists of included articles will be conducted. Two investigators will independently screen titles and abstracts for potential eligibility. Studies which report prevalence of diabetic retinopathy among general populations of children and young people with types 1 or 2 diabetes will be included. Pooled prevalence estimates of diabetic retinopathy reported in studies with sample size greater than 200 participants will be calculated by the random effect model. Forest plots will be used to summarise individual and pooled estimates of the prevalence. Heterogeneity between studies will be assessed using the $I^2$ statistic and explored through meta-regressions and subgroup analyses if the necessary data are available.

**Ethics and dissemination** Ethics approval is not required as this is a review of anonymised published data. We will report the findings of this systematic review in a peer-reviewed journal, and share it with the relevant professionals including health authorities through our Diabetic Eye disease in Childhood Study collaborative network.

**Clinical trail registration** PROSPERO (CRD42017067178).

### Strengths and limitations of this study

► This formal systematic review will provide currently unavailable information on the prevalence of diabetic retinopathy in the paediatric population.
► Only studies reporting diagnostic methods and grades of diabetic retinopathy will be included to enhance precision.
► To increase the applicability of the findings, two severity cut-offs will be used to calculate prevalence estimates.
► As with all systematic reviews, the poor reporting of methods in primary studies might compromise the exploration of the heterogeneity.

a significant increase in the number of children and young people at risk of visual loss.

Microvascular retinal damage is a marker of general vascular health, indicative of a need for greater diabetic disease control. The more advanced, but still asymptomatic, stages of diabetic retinopathy require additional ophthalmic intervention to reduce the risk of irreversible visual loss. Screening for diabetic retinopathy in children and young people is recommended by many national population health programmes, with commencement of annual retinal examination determined either by age (typically age 10–12 years) or duration of disease (typically 2–5 years from diagnosis).[5–7]

Although it has been suggested that these recommendations need to be revised based on recent evidence on the burden of the disease in this population, there is no consensus on the direction of these modifications.[8–12] Variable estimates of the prevalence of diabetic retinopathy may be due in part to heterogeneity between study populations, diagnostic methods or definitions of diabetic retinopathy used in the studies.[13 14] Consensus about screening strategies for the

## BACKGROUND

Diabetic retinopathy is the most common preventable cause of visual impairment in young adults.[1] The rising prevalence of childhood type 1 and type 2 diabetes[2–4] has led to

paediatric population required greater clarity in evidence base regarding the natural history and the burden of disease in this group.[15–17]

The aim of this systematic review is to estimate the prevalence of diabetic retinopathy in children and young people living with type 1 or type 2 diabetes, and to investigate potential sources of heterogeneity such as study sample characteristics (eg, age, diabetes type and duration, and glycaemic control), diagnostic methods and definition of diabetic retinopathy to inform screening strategies for the paediatric population.

## METHODS/DESIGN

The systematic review will be performed in accordance with the Preferred Reporting Items for Systematic Reviews and Meta-Analyses (PRISMA) guidelines, with this protocol written according to the PRISMA-P checklist.[18] The PRISMA-P checklist can be found as online supplementary additional file 1. The study has been registered on PROSPERO (CRD42017067178).

### Data sources and searches

The PubMed and EMBASE databases will be searched using the OvidSP platform. We will also manually review the references lists of identified articles. The search strategy (table 1) will include terms related to children or young people, diabetes, diabetic retinopathy, prevalence and diagnostic tests, and there will be no language restrictions. We will exclude studies published before January 1995 as older studies are likely to have been conducted prior to the publication of the Diabetes Control and Complications Trial, which demonstrated decreased risk of diabetic retinopathy development or progression with tighter glycaemic control than that previously used in routine clinical practice, and may have led to significant changes in disease management and natural history.[19]

### Study eligibility criteria

We will include studies which report prevalence estimates of diabetic retinopathy among general populations of children and young people with diabetes mellitus, or studies from which the prevalence data may be derived (table 2). Eligible studies will be those which meet the following criteria.

► Participants
  Individuals aged under 18 years living with types 1 or 2 diabetes. Studies with median/mean age of the sample greater than 25 years or which include participants older than 30 years will be excluded.
► Disease measure
  Report of the clinical examination method used to determine diabetic retinopathy status, and description of either retinal findings or grade of retinopathy using any grading systems which can be correlated to retinopathy features seen at the ophthalmic examination. This will allow comparison between the different grading systems used in screening programmes and

**Table 1** Search strategy MEDLINE (OvidSP)

| Category | # | Keyword |
|---|---|---|
| Children and young people | 1. | exp Child/ |
| | 2. | exp Adolescent/ |
| | 3. | exp Pediatrics/ |
| | 4. | child*.mp. |
| | 5. | teen*.mp. |
| | 6. | school age.mp. |
| | 7. | 1/OR 6 |
| Diabetes | 8. | exp Diabetes Mellitus, Type 1/ |
| | 9. | exp Diabetes Mellitus, Type 2/ |
| | 10. | exp Diabetes Complications/ |
| | 11. | diabet*.mp. |
| | 12. | 8 OR/11 |
| Retinopathy | 13. | exp Eye Diseases/ |
| | 14. | exp Diabetic Retinopathy/ |
| | 15. | exp Retinal Neovascularization/ |
| | 16. | exp Macular Oedema/ |
| | 17. | retinopath*.mp. |
| | 18. | maculopath*.mp. |
| | 19. | background retinopathy.mp. |
| | 20. | microaneurysm*.mp. |
| | 21. | 13 OR/20 |
| Prevalence | 22. | exp Epidemiology/ |
| | 23. | exp Prevalence/ |
| | 24. | exp Incidence/ |
| | 25. | exp Natural History/ |
| | 26. | frequenc*.mp. |
| | 27. | epidemiolog*.mp. |
| | 28. | prevalen*.mp. |
| | 29. | inciden*.mp. |
| | 30. | severity.mp. |
| | 31. | 22 OR/30 |
| Diagnostic tests | 32. | exp Photography/ |
| | 33. | exp Ophthalmoscopes/ |
| | 34. | exp Fluorescein Angiography/ |
| | 35. | exp Fundus Oculi/ |
| | 36. | exp Slit Lamp/ |
| | 37. | exp ophthalmoscopy/ |
| | 38. | Tomography, Optical Coherence/ |
| | 39. | Ultrasonography/ |
| | 40. | photo*.mp. |
| | 41. | ophthalmoscop*. mp. |
| | 42. | angiogram*. mp. |
| | 43. | biomicroscopy.mp. |
| | 44. | Optical coherence tomography.mp. |
| | 45. | ultrasonograph*.mp. |
| | 46. | fundus.mp. |
| | 47. | Slim Lamp.mp. |
| | 48. | fundoscop*.mp. |
| | 49. | 32 OR/48 |
| | 50. | 7 AND 12 AND 21 AND 31 AND 49 |
| | 51. | limit 50 to yr='1995 –Current' |

**Table 2** Inclusion and exclusion criteria

| Category | Inclusion criteria | Exclusion criteria |
|---|---|---|
| Population | ► Children and young people <18 years old included in the sample<br>► Children and young people from general population | ► Median/mean sample age >25 years<br>► Participants >30 years old included<br>► Studies conducted in selected populations (eg, dialysis, post-transplanted) |
| Exposure | ► Type 1 or 2 diabetes | ► Other types of diabetes (eg, monogenic causes) |
| Outcome of interest | ► Prevalence of diabetic retinopathy<br>► Test used to assess diabetic retinopathy reported | ► Estimation of prevalence not possible with published data |
| Article | ► Published between 1 January 1995 and 31 October 2016 | ► Duplicate reports (the most comprehensive version will be included) |

clinical practice. Included grading systems will be displayed in a supplementary table along with the systematic review's results.

Prevalence will be calculated using two cut-offs, *any* diabetic retinopathy and s*ight-threatening* diabetic retinopathy, that is, grades of retinopathy that require prompt ophthalmic management to reduce the risk of irreversible visual impairment. *Any* diabetic retinopathy will be defined as background retinopathy or more severe disease. *Sight-threatening* diabetic retinopathy[20] will be primarily defined as proliferative diabetic retinopathy or *clinically significant* maculopathy oedema, that is, retinal thickening at or within 500 μm of the centre of the macula, and/or hard exudates at or within 500 μm of the centre of the macula, if associated with thickening of the adjacent retina, and/or a zone or zones of retinal thickening one disc area in size, any part of which is within one disc diameter of the centre of the macula.[21] Additionally, a secondary analysis using a more inclusive definition of sight-threatening diabetic retinopathy, that is, including severe non-proliferative diabetic retinopathy, will be conducted using the available data.

Conference abstracts with sufficient data on disease measure to determine eligibility will be included.

## Study selection
The search results will be extracted and managed using Microsoft Excel (Microsoft Corporation, Redmond, Washington, USA). Two independent investigators (MIB, ALS) will screen titles and abstracts, and then apply the eligibility criteria to full-text articles. Disagreements will be resolved by consensus following discussion. Arbitration by a third author (JSR) will be sought whenever necessary.

## Data extraction
Two investigators (MIB, ALS) will extract the data using a standardised form. The following data items will be extracted: first author, year of publication, design, country, year of study conduction, sample size, diabetes type, age at examination, diabetes duration, HbA1c, diagnostic test, diabetic retinopathy definition, number of any diabetic retinopathy cases, severity of diabetic retinopathy cases. If a study reports multiple prevalence estimates over time,

the first prevalence estimate, that is, the estimate which is reported with baseline sample clinical characteristics, will be selected as the most informative measure. Characteristics of children and young people with sight-threatening diabetic retinopathy will be also extracted. Authors will be contacted if data in the original publication is unclear.

## Assessment of risk of bias in included studies
The risk of bias will be assessed with an adapted version of the tool developed by Hoy and colleagues[22] (online Supplementary additional file 2). The sample will be considered representative of the target population (ie, 'general' population) when all children and young people registered in a paediatric diabetes clinic, or living in a specified catchment area were potentially eligible for the study. Sources of information will be classified as appropriate where diabetic retinopathy status is assessed during the study (cross-sectional and prospective studies) or where prospective databases such as diabetic retinopathy screening registers are used.

## Data synthesis
A comprehensive table for summary of findings with narrative description of all eligible studies will be reported. However, only studies with a sample size equal or greater than 200 participants will be included in the meta-analysis. Small studies will be excluded to reduce variability due to imprecise estimates. Pooled prevalence estimates of any diabetic retinopathy and of sight-threatening diabetic retinopathy will be calculated by random effect model meta-analysis. Forest plots will be used to summarise individual and pooled estimates of the prevalence. Heterogeneity between studies will be assessed using the $I^2$ statistic (values of 25%, 50% and 75% will be considered low, medium and high heterogeneity, respectively)[23] and will be further explored through meta-regressions and subgroup analyses considering the following covariates: sample median/mean age, median/mean diabetes duration, type of diabetes, median/mean HbA1c, diagnostic test, and year of study conduction. All analyses will be performed with *R* statistical software V.3.3.2., and p values<0.05 will

 

be used as the threshold for statistical significance. Assessment of meta-bias(es) is not planned, as preliminary searches of registered protocols have indicated that it is unlikely that eligible epidemiological studies are prospectively registered. As this systematic review is not investigating the evidence underpinning an intervention, confidence in the cumulative evidence cannot be calculated or assessed using the Grading of Recommendations Assessment, Development and Evaluation. The strength of the body of evidence will be examined through the assessment of risk of bias within included studies.

## DISCUSSION

Diabetic retinopathy is, in the early stages, an important marker of disease control, and in the later stages, a preventable cause of visual impairment and blindness. The burden of childhood diabetes is increasing, but there is a paucity of synthesised data on the prevalence and natural history of childhood diabetic retinopathy. This systematic review, using a clinically impactful taxonomy to calculate the prevalence of diabetic retinopathy, will provide currently unavailable prevalence estimates to inform screening policy and practice for children and young people living with diabetes.

### Ethics and dissemination

Ethical approval is not needed as this study will use only anonymised published data. We will report the findings of this systematic review in a peer-reviewed journal, and share it with the relevant professionals including health service planners through our DECS collaborative network.

**Contributors** JSR is the guarantor. MCIB designed the study and drafted the protocol. ALS, PMC and JSR critically revised and provided feedback on the protocol. MCIB, ALS, PMC and JSR read and approved the final version of the manuscript.

**Funding** This work was funded by the Ulverscroft Foundation, the National Institute for Health Research (NIHR) Biomedical Research Centre at University College London Institute of Child Health/Great Ormond Street Hospital NHS Foundation Trust, Diabetes Research & Wellness Foundation and the National Commission for Scientific and Technological Research in Chile (CONICYT). ALS and JR are also funded by the NIHR Moorfields Biomedical Research Centre. The funding organisations had no role in the design or conduct of this research.

**Competing interests** None declared.

**Provenance and peer review** Not commissioned; externally peer reviewed.

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
