## [Reviewer comments · BMJ Open]

ARTICLE DETAILS

TITLE (PROVISIONAL)	Prevalence of diabetic retinopathy in children and young people living with diabetes: protocol for a systematic review.
AUTHORS	Ibanez-Bruron, Maria; Solebo, AL; Cumberland, Phillipa; Rahi, Jugnoo

VERSION 1 – REVIEW

REVIEWER	PedroRomero-Aroca Ophthalmology Service, University Hospital Sant Joan, Institut de Investigacio Sanitaria Pere Virgili [IISPV], Universitat Rovira & Virgili, Reus [Spain].
REVIEW RETURNED	13-Jul-2017

GENERAL COMMENTS	A. Summary Manuscript that proposed how can be made a protocol for diabetic retinopathy prevalence in children. B. Strengths: Clearly defined technique to elaborate a protocol to determine diabetic retinopathy after systematic review of published studies. C. Weakness. Weaknesses of this study are: 1. Authors included only papers that aged children will be under 18 years old, that only permit to know the prevalence of diabetic retinopathy (DR) in people with less than 18 years old, is known that DR appeared more frequently after this age.2. Authors only include articles that uses ETDRS diabetic retinopathy diagnosis technique, currently ETDRS seven fields retinographies is not used in clinical practice, usually screening techniques uses one field retinography and for diagnosis uses two (EURODIAB) or three (JOSLIN protocol) 45° field retinographies, it will be difficult to encounter articles that only uses ETDRS technique. Authors said in same paragraph that equivalent ETDRS technique articles will be studied, it is necessary to describe which retinography protocol could be used.3. Sight threatening diabetic retinopathy (STDR) include patients with only clinical significant diabetic macular edema or proliferative DR, but with this diagnosis criteria a lot of people with macular ischemia due to severe or moderate diabetic retinopathy will be excluded, it is an error. Usually STDR is defined as = level 43 or worse as defined by the ETDRS classification and should be include patients with ischemic maculopathy.
---

REVIEWER	Stuart Keel The Centre for Eye Research Australia No Competing Interest
REVIEW RETURNED	12-Sep-2017

GENERAL COMMENTS	Thank you for the opportunity to review this protocol. There is clear justification for the review to be conducted and the methodology is sound. I just have a few minor comments;  - P 3. Line 37. Regarding 'pooled prevalence' – Further justification is warranted as to the appropriateness of pooling the results of studies that have utilized different methodologies. E.g. hospital based vs. population-based. - P 7. Line 44. Please define clinically significant macular edema. I.e. when oedema was within 500 microns of the foveal centre +/- if focal photocoagulation scars were present in the macular area. - Sight-threatening DR usually also includes severe NPDR. What will you do if grading schemes are not compatible across studies? - P 8. Line 34. Regarding the statement - 'If a study reports multiple prevalence estimates over time, the first prevalence estimate will be selected.' - It is unclear as to why you will not use the most recent prevalence estimate as the reference point.
--

VERSION 1 – AUTHOR RESPONSE

Reviewer Name: Pedro Romero-Aroca

A. Summary

Manuscript that proposed how can be made a protocol for diabetic retinopathy prevalence in children.

B. Strengths:

Clearly defined technique to elaborate a protocol to determine diabetic retinopathy after systematic review of published studies.

C. Weakness.

Weaknesses of this study are:

1. Authors included only papers that aged children will be under 18 years old, that only permit to known the prevalence of diabetic retinopathy (DR) in people with less than 18 years old, is known that DR appeared more frequently after this age.

Response: As the reviewer noted, it is known that diabetic retinopathy appears more frequently after childhood. However, the purpose of this study is to estimate the prevalence of diabetic retinopathy in children and young people living with diabetes so to inform screening recommendations for this group. Prevalence estimates from adult samples could not be directly applied to our central question.

2. Authors only include articles that uses ETDRS diabetic retinopathy diagnosis technique, currently ETDRS seven fields retinographies is not used in clinical practice, usually screening techniques uses one field retinography and for diagnosis uses two (EURODIAB) or three (JOSLIN protocol) 45° field retinographies, it will be difficult to encounter articles that only uses ETDRS technique. Authors said in same paragraph that equivalent ETDRS technique articles will be studied, it is necessary to describe which retinography protocol could be used.

Response: We will include any grading systems which can be correlated to retinopathy features seen at the ophthalmic examination. This will allow comparison between the different grading systems used in screening programmes and clinical practice. Included grading systems will be displayed in a supplementary table along with the systematic review's results.

3. Sight threatening diabetic retinopathy (STDR) include patients with only clinical significant diabetic macular edema or proliferative DR, but with this diagnosis criteria a lot of people with macular ischemia due to severe or moderate diabetic retinopathy will be excluded, it is an error. Usually STDR is defined as = level 43 or worse as defined by the ETDRS classification and should be include patients with ischemic maculopathy.

Response: As the reviewer noted in his previous comment, currently ETDRS seven fields retinographies is not commonly used in clinical practice or screening programmes. The majority of the grading systems only consider 4 or 5 categories, without always differentiating moderate from severe non-proliferative diabetic retinopathy (e.g. the English National Screening Protocol). The most important cut-offs, that are shared by most of the grading systems and determine changes in management, are any diabetic retinopathy and sight threatening diabetic retinopathy defined as proliferative disease or diabetic macular edema. A secondary analysis using a more sensitive definition of sight threatening diabetic retinopathy, i.e. including severe non-proliferative diabetic retinopathy, will be conducted using the available data. Additionally, as the reviewer will be aware, macular ischemia is rare in the paediatric population so it is unlikely to be a major issue.

Reviewer name: Stuart Keel

Thank you for the opportunity to review this protocol. There is clear justification for the review to be conducted and the methodology is sound. I just have a few minor comments;

Comment: P 3. Line 37. Regarding 'pooled prevalence' – Further justification is warranted as to the appropriateness of pooling the results of studies that have utilized different methodologies. E.g. hospital based vs. population-based.

Response: We believe that the inclusion criteria of the systematic review are strict enough to ensure the comparability. However, as explained in the protocol, heterogeneity between studies, for example whether study populations are hospital or population based, will be assessed using the I² statistic and will be further explored through meta-regressions and sub-group analyses. Additionally, random effect meta-analyses will be used when studies displayed high heterogeneity.

Comment: P 7. Line 44. Please define clinically significant macular edema. I.e. when oedema was within 500 microns of the foveal centre +/- if focal photocoagulation scars were present in the macular area.

Response: We have amended the paper to describe clinically significant macular edema using an internationally agreed taxonomy (ETDRS), i.e. retinal thickening at or within 500 µm of the center of the macula; and/or hard exudates at or within 500 µm of the center of the macula, if associated with thickening of the adjacent retina; and/or a zone or zones of retinal thickening one disc area in size, any part of which is within one disc diameter of the center of the macula.

Comment: Sight-threatening DR usually also includes severe NPDR. What will you do if grading schemes are not compatible across studies?

Response: As the reviewer noted, currently more than one grading system are being used, which are not always compatible. So we will extract, along with the number of cases, the grading system and the severity of cases.

The majority of the grading systems only consider 4 or 5 categories, without always differentiating moderate and severe non-proliferative diabetic retinopathy (e.g. the English National Screening Protocol). The most important cut-offs, that are common to most of the grading systems and determine changes in the management of patients, are any diabetic retinopathy and sight threatening diabetic retinopathy (STDR) defined as proliferative disease or diabetic macular edema. A secondary analysis using a more sensitive definition of sight threatening diabetic retinopathy (STDR), i.e. including severe non-proliferative diabetic retinopathy, will be conducted using the available data.

Comment: P 8. Line 34. Regarding the statement - 'If a study reports multiple prevalence estimates over time, the first prevalence estimate will be selected.' - It is unclear as to why you will not use the most recent prevalence estimate as the reference point.

Response: We will select the most informative prevalence estimate, i.e. the estimate that can be correlated to more sources of heterogeneity. The first prevalence is usually the most informative as it is reported with baseline sample characteristics, e.g. age, diabetes duration and control.